# Characterization of Mycoviruses in *Armillaria ostoyae* and *A. cepistipes* in the Czech Republic

**DOI:** 10.3390/v16040610

**Published:** 2024-04-15

**Authors:** Lucie Walterová, Leticia Botella, Ondřej Hejna, Marcos de la Peña, Tomáš Tonka, Vladislav Čurn

**Affiliations:** 1Department of Genetics and Agricultural Biotechnology, Faculty of Agriculture and Technology, University of South Bohemia in České Budějovice, Na Sádkách 1780, 370 05 České Budějovice, Czech Republic; hejna@fzt.jcu.cz (O.H.); ttonka@fzt.jcu.cz (T.T.); curn@fzt.jcu.cz (V.Č.); 2Department of Forest Protection and Wildlife Management, Faculty of Forestry and Wood Technology, Mendel University in Brno, Zemědělská 1, 613 00 Brno, Czech Republic; leticia.sanchez@mendelu.cz; 3Instituto de Biología Molecular y Celular de Plantas, Universidad Politécnica de Valencia-CSIC, 46022 Valencia, Spain; rivero@ibmcp.upv.es

**Keywords:** root rot, viruses, circular genetic elements, ambivirus, tymovirus, biological control

## Abstract

Members of the genus *Armillaria* are widespread forest pathogens against which effective protection has not yet been developed. Due to their longevity and the creation of large-scale cloning of *Armillaria* individuals, the use of mycoviruses as biocontrol agents (BCAs) against these pathogens could be an effective alternative. This work describes the detection and characterization of viruses in *Armillaria* spp. collected in the Czech Republic through the application of stranded total RNA sequencing. A total of five single-stranded RNA viruses were detected in *Armillaria ostoyae* and *A. cepistipes*, including viruses of the family *Tymoviridae* and four viruses belonging to the recently described “ambivirus” group with a circular ambisense genome arrangement. Both hammerhead (HHRz) and hairpin (HpRz) ribozymes were detected in all the ambiviricot sequences. *Armillaria* viruses were compared through phylogenetic analysis and confirmed their specific host by direct RT-PCR. One virus appears to infect both *Armillaria* species, suggesting the occurrence of interspecies transmission in nature.

## 1. Introduction

Fungal pathogens are important members in forest ecosystems. They affect the diversity, structure and dynamics of forest communities substantially [1,2]. The fungal genus *Armillaria* includes more than 40 described species [3], which are causal agents of root rot in a wide variety of trees, shrubs and some herbs [4], including economically important conifers (e.g., *Abies*, *Picea*, *Pinus*) and agricultural crops (e.g., *Citrus*, *Juglans*, *Malus*, *Prunus*, *Vitis*) [5]. Norway spruce (*Picea abies*) is a major host of *Armillaria* in the Czech Republic. Over the last few years, Czech forests of Norway spruce seem to be suffering a generalized decline [6,7], which is enhanced by pathogenic fungi like *Armillaria* [8,9]. The decline is often caused by a combination of abiotic and biotic factors that limit tree growth, reduce foliage quality, and weaken root systems. These interactions can kill individual trees and entire stands [10,11]. Although there are relatively efficient methods for the protection of agricultural crops against *Armillaria* root rot, none of the established methods can completely eliminate the mycelium from the contaminated site [12,13]. Therefore, other strategies have to be pushed towards the development of environmentally friendly control approaches [14].

Fungal viruses (mycoviruses) of plant-interacting fungi are particularly significant for crop protection because they can influence the phenotype of their host. An increasing number of mycoviruses with the ability to induce hypovirulence in their host have been reported [15,16,17]. There are important forest pathogenic fungi, such as *Cryphonectria parasitica*, *Phytophthora* spp., *Ophiostoma* spp., *Gremmeniella abietina*, *Hymenoscyphus fraxineus*, *Heterobasidion annosum* and *Armillaria* spp., which host mycoviruses with diverse fungus-virus relationships [18,19,20]. In *Armillaria*, early studies by Blattny (1973) [21] and Reaves et al. (1988) [22] described the presence of virus-like particles in *A. mellea* and *A. ostoyae*, but it was not until 2021 that viruses were confirmed by molecular characterization [13,23,24]. 

Previous research on Armillaria root rot fungi has identified several RNA viruses. These studies have primarily aimed to understand the diversity and impact of these viruses on *Armillaria* species, focusing on their ecological roles and potential effects on fungal virulence. To date, the virome of the genus Armillaria has been molecularly investigated in samples from Finland, Russia (Siberia), South Africa and Switzerland. Several positive- and negative-stranded ssRNA viruses and even two dsRNA mycoviruses have been described. Mitoviruses, ourmia-like viruses, tymoviruses, phlegiviruses, ambi-like viruses, several previously unclassified (+)ssRNA and two unclassified dsRNA viruses have been described in five different *Armillaria* species (*A. mellea*, *A. borealis*, *A. cepistipes*, *A. ostoyae*, *A. gallica*) [13,23,24].

The main objectives of this study were (i) to confirm the occurrence of mycoviruses in a collection of isolates of the genus *Armillaria* in Central Europe, and (ii) to describe their genomic features and phylogenetic relationships.

## 2. Materials and Methods

### 2.1. Sampling

*Armillaria* isolates were collected in the years 2019–2020 in the Czech Republic, mainly in northeastern Moravia, the Moravian Karst and southeastern Bohemia (Table 1). Spores from fruiting bodies or rhizomorphs were transferred and cultured on ME agar (7.5 g bacteriological agar, 3.5 g corn agar, 10 g malt extract, 10 g glucose/0.5 L; pH 5.5) at 25 °C for approximately four weeks before being harvested. Subsequently, samples on the same ME agar were stored at 4 °C. A total of 13 samples, including *Armillaria ostoyae* and *A. cepistipes*, were cultivated and analyzed. 

### 2.2. Isolation of Double-Stranded (ds) RNA

The samples were first examined for the presence of potential viral dsRNA molecules. Mycelium from one Petri dish was lyophilized and collected in a 50 mL Falcon tube. Two stainless steel beads with a diameter of 10 mm and liquid nitrogen were added to the Falcon tube and mycelium was homogenized by vortexing at maximum speed (3400 RPM/approx. 2 min). After homogenization, dsRNA was extracted using protocol Morris and Dodds (1979) [25] with some modifications, as described in the work of Tonka et al. (2021) [26]. Previously confirmed dsRNA-hosting isolates of *Gremmeniella* [27] and *Phytophthora* [28] were used as the positive control of each isolation.

### 2.3. Isolation of Total RNA

Thirteen samples were selected for stranded total RNA sequencing analysis (Table 1). Approximately 50–100 mg of fresh mycelium was collected from the agar of each sample and transferred to the 2 mL tubes with steel beads. The tubes were immersed in liquid nitrogen and vortexed (using Labnet vortex mixer VX-200, Labnet, Edion, NJ, USA) until the mycelium was ground to a fine powder. Isolation of total RNA was performed using a SPLIT RNA Extraction Kit (Lexogen, Vienna, Austria) following the protocol provided by the manufacturer, eluted in 30 µL of EB buffer. Total RNA was visualized on an agarose gel and stored at −80 °C.

### 2.4. Stranded Total RNA Sequencing

The isolated total RNA of 13 *Armillaria* samples was pooled into one mixed sample and treated with TURBO DNA-free TM Kit (Invitrogen^TM^ (Waltham, MA, USA), Thermo Fisher Scientific (Waltham, MA, USA)). RNA quantity was measured by Qubit^®^ Fluorometer (Life Technologies (Carlsbad, CA, USA), Thermo Fisher Scientific). Total RNA was sent to SEQme s.r.o. (Dobris, Czech Republic) for RNA library construction and RNA sequencing. Prior to the library preparation, ribosomal RNA (rRNA) was depleted using the NEBNext rRNA Depletion Kit (Human/Mouse/Rat). The library was prepared using an NEBNext Ultra II Directional RNA Library Prep Kit for Illumina. The quality of the prepared library was checked using an Agilent Bioanalyzer 2100 High sensitivity DNA Kit, Invitrogen Collibri Library Quantification Kit and Qubit 1X dsDNA High-Sensitivity Assay Kit. A KAPA Library Quantification Kit for Illumina platform was used for absolute qPCR-based quantification of the Illumina libraries flanked by the P5 and P7 flow cell oligo sequences. Libraries underwent paired-end (PE) (2 × 150 nt) sequencing on a NovaSeq6000 (DS-150) (Illumina, San Diego, CA, USA) using a NovaSeq S4 v1.5 reagent kit. An “in-lane” PhiX control spike was included in each lane of the flow cell.

### 2.5. Bioinformatics

#### 2.5.1. Data Preprocessing

The raw sequencing data were downloaded from the data repository of the sequencing company SEQme s. r. o. (Dobris, Czech Republic). The processing was carried out on the local server of the University of South Bohemia. In the first step, data quality was assessed using the FASTQC v.0.11.9 program [29]. This control step revealed the presence of a small number of adapter sequences. The nucleotide sequences of adapters were obtained according to the used library kit and sequencer machine mentioned earlier [30]. Adapter and quality trimming were performed using the program Cutadapt v3.4 [31] with requirements for a Phred score higher than 30 and a minimum length of the truncated sequence of 50 bases. The software is available at the code depository GitHub [32]. Subsequently, data quality of trimmed reads was again assessed in FASTQC program mentioned previously.

#### 2.5.2. Host Reads’ Removal

After preprocessing the data, it was necessary to remove reads belonging to host fungi. The STAR v2.7.9a program was used for this purpose [33]. Assembly GCA_900157425.1 version 2 of strain C18/9 of *A. ostoyae* was used as a reference sequence. After mapping the reads to the *Armillaria* genome, only unmapped reads were left for further processing.

#### 2.5.3. Discovery of Known Virus 

To reveal already known viruses, the Viral NCBI database [34] was used for this purpose. We downloaded only complete viral RefSeq nucleotide sequences. Our reads were aligned to this reference database by BWA v0.7.17.-r1188 program package [35]. In the next step, we calculated the coverage for each viral genome using Samtools kit v1.16.1 [32]. If the coverage was higher than 80 percent, we visualized the alignment for particular viral genome with IGV program v2.16.1 [36] and manually confirmed or denied the presence of the published virus.

#### 2.5.4. Discovery of Novel Viruses 

In the first step, de novo assembly was performed with unmapped reads from STAR aligner. The toolkit SPAdes version 3.15.3 [37] with default settings for metagenomics was used for this purpose. Assembled contigs shorter than 1000 bp were discarded. In the next step, the rest of contigs were compared to several databases: Viral UniProt KB [38], Viral NCBI [34], RVDB [39], Virus-Host DB [40]. Each of the databases was downloaded to a local server and was used to search for similarities between database sequences and assembled contigs. For this purpose, the program BLASTx and BLASTn v2.12.0+ was used. The threshold for the E-value was set to 1 × 10^−3^. All contigs with at least one hit were further used to search for similarity with the NCBI database nr (RefSeq non-redundant proteins) or nt (RefSeq Nucleotide) specifying the *Armillaria* taxon. If a positive hit was found, the E-value values were subsequently compared. If this value was higher for the results from viral databases, the contig was removed. In the last step, the remaining contigs were searched with the entire nr and nt databases and the threshold for the E-value was set as 1 × 10^−5^. 

Potential protein-encoding segments were detected with a coding open reading frame (ORF) finder using Geneious^®^ v.-8.1.9. Depth of coverage: for the calculation of the coverage depth, we used the following formula: (Total reads mapped to the final identified virus * average read length)/virus genome or contig length.

#### 2.5.5. Ribozymes’ Detection

The identification of potential ribozymes followed a customized protocol adapted from the viroid-like sequence search pipeline outlined by Lee et al. in 2023 [41]. The process began with the recognition of covalently closed circular RNAs (cccRNAs) using a modified version of the reference-free CIRIT algorithm, as introduced by Qin et al. in 2020 [42]. This algorithm entails the exploration of overlapping regions between the starting and ending points of contigs to pinpoint cccRNAs. Unlike the original algorithm, our modified version repeatedly attempted to dissect potential cccRNAs into individual units, with the condition of maintaining a minimum 95% similarity within repeated regions. Subsequently, the identified cccRNAs underwent a search for known self-cleaving ribozymes, employing Infernal, a tool introduced by Nawrocki and Eddy in 2013 [43]. Ribozymes surpassing Rfam’s curated gathering threshold or exhibiting E-values below 0.1 were identified in each polarity. In addition to this, a subset of sequences underwent further scrutiny using an RNA motif, a tool developed by Macke et al. in 2001 [44]. This supplementary analysis aimed to uncover more diverse ribozymes that may not have been initially detected by Infernal. 

### 2.6. Retro Transcription Polymerase Chain Reaction (RT-PCR) and Sanger Sequencing

Approximately 50–100 mg of fresh mycelium was collected and crushed in a mortar. Crushed mycelium was mixed with 600 µL of lysis buffer (LB; included in the PureLinkTM RNA Mini Kit) and 6 µL 2-mercaptoethanol and transferred to the 1.5 mL sterile tube. Isolation of total RNA was carried out using PureLinkTM RNA Mini Kit (Invitrogen, USA). The RNA was eluted in 30 μL of elution buffer and 1 µL of RNase inhibitor was added. The quality of total RNA was assessed by gel electrophoresis in 1.5% agarose gel and total RNA was stored at −80 °C. LunaScript^®^ RT SuperMix Kit (New England Biolabs, Hitchin, UK) was used for the synthesis of cDNA, as reported by Tonka (2022) [45]. The success of cDNA synthesis was verified by amplification of the eukaryotic region translation elongation factor 1-alpha-eEF1A (tefa) and checked by gel electrophoresis in 1.5% agarose gel [46]. If the amplification was successful, the cDNA was used in a PCR reaction with virus-specific primers. The virus-specific primers were designed to partially amplify the ORFs of putative viruses based on sequencing data from RNA sequencing using program Geneious v.-8.1.9. PCR amplification was performed with 12 μL PPP Master Mix (Top-Bio, Vestec, Czech Republic) 1 μL of each 10 mM primer (Appendix A), 4 μL of cDNA and PCR-grade water in a total volume of 25 μL. PCR conditions were as follows: 94 °C 2 min, 25× (94 °C 1 min, 60 °C 1 min, 72 °C 2 min) 72 °C 5 min. Due to the low concentration of putative viruses, the PCR amplification was repeated with fresh chemicals and 4 μL of resulting product from the first PCR amplification was used as a template; then, PCR products were visualized by gel electrophoresis in 1.5% agarose gel. All resulting fragments with the appropriate size were cut from the gel, cleaned using NucleoSpin^®^ Gel and PCR Clean-up kit-Macherey-Nagel (BioTech a.s, Prague, Czech Republic), and ExoSAP-IT™ PCR Product Cleanup Reagent (Thermo Fisher Scientific, USA), and then fragments were sent to SEQme (Dobříš, Czech Republic) for Sanger sequencing. 

### 2.7. Conserved Domains

NCBI CD-search tool [47] was used to search for putative conserved domains in predicted amino acid (aa) sequences. All sequences were then aligned to aa sequences of related viruses retrieved from the GenBank using MUSCLE v3.8.425 in Geneious v.-8.1.9.

### 2.8. Phylogenetic Analyses

Amino acid (aa) sequences of RdRP regions of each virus were included in phylogenetic analyses. Sequences were aligned in Geneious v. 8.1.9 by MUSCLE v3.8.425 [48] together with known aa sequences of viruses considered to be related. Phylogenetic trees were built using the maximum likelihood method [49] in RAxML-HPC v.8 on XSEDE running in the CIPRES Science Gateway web portal [50]. Bootstrapping was performed by using the recommended parameters provided by the CIPRES Science Gateway portal. The trees were visualized in FIGTREE (V1.4.4). 

## 3. Results and Discussion

In this study, we described the genomes of five putative single-stranded (ss) RNA viruses hosted by pathogenic fungi belonging to the genus *Armillaria* (specifically *A. ostoyae* and *A. cepistipes*) from the Czech Republic. These new viruses include a member of the family *Tymoviridae* and members of recently described virus group, tentatively named “Ambiviruses” [51,52,53,54].

### 3.1. dsRNA Screening

No dsRNA elements were detected in Czech isolates of *Armillaria* spp. As later demonstrated by high-throughput sequencing (HTS) of total RNA, no dsRNA viruses appear to occur in our collection of isolates. Some ssRNA viruses can produce dsRNA intermediates during their replication. However, not all ssRNA viruses generate dsRNA, and the amount produced varies among viral species. While dsRNA extraction methods can capture dsRNA, they may not effectively detect all ssRNA viruses, especially those with low dsRNA abundance. Our result agrees with that of the study of Linnakoski et al. (2021) [13], which dealt with the detection of mycoviruses in isolates of fungi of the genus *Armillaria* from Finland, Russia, and North Africa, and it also corresponds to the results of Dvořák’s work (2008) [55], which did not detect dsRNA viruses in *Armillaria* from the Czech Republic. However, Shamsi et al. (2024), in their work, report two detected mycoviruses with a dsRNA genome (a partitivirus and a phlegivirus) in samples originating from Switzerland [24]. Dvořák, in 2008 [55], and Blattný et al., in 1973 [21], studied the *Armillaria* fungi virome from the Czech Republic in their works. In his research, Dvořák (2008) [55] detected possible double-stranded molecules in forty samples from the Czech Republic, but they were not confirmed to be viral, which coincides with the results of this study. Likewise, Blattný et al. (1973) [21] and Reaves et al. (1988) [22] detected virus-like particles in their works on *Armillaria mellea* and *Armillaria ostoyae*, but they were not further verified molecularly. Blattný et al. (1973) [21] described these particles as rod-shaped (22–28 × 119 nm) or isometric (30 nm).

### 3.2. Identification of ssRNA Viruses

Total RNA sequencing generated 113 million paired-end (PE) reads. After quality trimming and de novo assembly, a total of 8809 contigs longer than 1 kb were obtained. BLASTX comparison of the contigs revealed four viral contigs with sequence similarities to members of the recently described circular RNA virus group “Ambiviruses” and one related to the members of the order *Tymovirales*, in particular, the family *Tymoviridae* (Table 2). 

Regarding ambi-like viruses, Sutela et al.’s work in 2020 [51] was the first study to describe “ambiviruses” in the endomycorrhizal fungi *Ceratobasidium* sp. and *Tulasnella* sp. Then, they were discovered in *Cryphonectria parasitica* [52] and many agaricomycetes including *Armillaria* spp. [13,24], *Heterobasidion* spp. [20], *Rhizoctonia* spp. [13], and *Phlebiopsis gigantea* [56]. As in these studies, the genomic sequences of the putative ambi-like viruses detected in Czech isolates of *Armillaria* spp. were approx. 4.5 kb long and contained typical RdRP conserved domains including the GDD motif (Appendix A). Based on the pairwise sequence comparison (PASC) percentages (Appendix A), four different viruses were identified and considered different species following the same criteria as those of Sutela et al. in 2020 [51], Forgia at al. in 2021 [52] and Turina et al. in 2023 [53]

They were tentatively named considering their host identity (Table 2). AALV1, AoALV2 and AoALV3 contained two and AoALV4 contained three non-overlapping ambisense ORFs. AALV1 encodes a first positive (+) sense ORF (ORF A) corresponding to the RdRP (713 aa long) and a second negative (−) sense ORF (ORF B) corresponding to a hypothetical protein (HP) 410 aa long. AoALV2 encodes the (−) sense ORF B-HP 655 aa long, and the (+) sense ORF A-RdRP 710 aa long. AoALV3 contains first (+) sense ORF corresponding to a hypothetical protein (ORF B) 400 aa long and a second negative sense ORF (ORF A) corresponding to RdRP 682 aa long. This arrangement shows that the contig was assembled in the 3′ to 5′ orientation. AoALV4 encodes a first (−) sense ORF B-HP 405 aa long, a second negative ORF 220 aa long, and a third (+) sense ORF A-RdRP 713 aa long. 

Ambivirus genomes have the unique feature of having circular genomes encoding RdRP and divergent ribozymes in various combinations in both sense and antisense orientations [54]. In our analyses, several hammerhead (HHRz) and hairpin (HpRz) ribozymes were detected. This is in agreement with the findings of Forgia et al. (2023) [54], who describe the HHRz and HpRz motifs and predicted cleavage sites in both known fungal ambiviruses and ambiviricot RdRP palmprints of 439 distinct species-like operational taxonomic units (sOTUs) found in GenBank. Although HHRz typically consists of 30 to 40 nucleotides (nts), in *Armillaria* ambiviruses, some of them seem to be slightly longer, ranging from 55 to 78 nts. The representation of *Armillaria* ambiviruses and their ribozymes is illustrated in Figure 1A–D, and the positions in the particular ambivirus contigs are shown in Appendix A.

The phylogenetic relationships of Czech *A. ostoyae* ambiviruses with other “ambiviruses” (Figure 2) show that they group in two separated virus clusters, both of them including viruses from *A. mellea* and *A. borealis*. This result suggests a monophyletic origin of the ambiviricot sequences in *Armillaria* spp. 

An Armillaria tymovirus 1. A viral contig resembling features of those from the genus *Tymoviridae* was detected in *Armillaria ostoyae*. Armillaria ostoyae tymovirus 1 is approximately 6.8 kb length and encodes one (+) sense ORF with 2172 aa (Figure 1E). Based on the sequence analysis of tymoviruses, conserved sequences were revealed (Appendix A), including conserved regions belonging to pfam01660 super family Vmethyltransf cl03298 in region 143–314 (E-value 6.72 × 10^−9^). This methyltransferase domain has been detected in a wide range of viruses and is involved in mRNA capping. Furthermore, conserved sequences were detected belonging to pfam01443 superfamily Viral_helicase1 cl26263 located 1113–1322 (E-value 2.66 × 10^−14^) and ps-ssRNAv_RdRp-like super family cl40470 located 1600–1799 (E-value 2.73 × 10^−25^) [57].

The order *Tymovirales* was first described in 2004 and currently comprises five families: *Alphaflexiviridae*, *Betaflexiviridae*, *Gammaflexiviridae*, *Deltaflexiviridae,* and *Tymoviridae* [58]. Members of the order *Tymovirales* have a 5.9 to 9.0 kb (+) ssRNA genome that is often polyadenylated. The largest protein, the replication-associated polyprotein (RP), is encoded by all members of the *Tymovirales* order. *Tymovirales* RP usually contains sets of conserved functional domains [59].

Viruses belonging to this order usually have a wide range of host organisms. *Betaflexiviridae* and *Tymoviridae* are usually considered plant viruses [59], a single member of *Gammaflexiviridae* has been detected in a filamentous fungus [58,60], and *Alphaflexiviridae* have been detected both in plants and fungi [61]. Although most viruses in the order *Tymovirales* are plant viruses, several members of this order are known to infect plant pathogenic fungi, including *Botrytis cinerea*, *Fusarium boothii*, *Fusarium graminearum*, *R. solani*, and *S. sclerotiorum* [62,63,64,65,66]. Armillaria ostoyae tymovirus 1 is the second tymovirus described within the genus *Armillaria* after the study of Shami et al. (2004) [24].

The phylogenetic relationships with other members from the order *Tymoviriales* from GenBank are shown in Figure 3. The phylogenetically closest virus to the virus AoTV1 is Lentinula edodes tymo-like virus 1, which was detected in a Chinese sample of the fungus *Lentinula edodes*.

### 3.3. RT-PCR Screening

Some of the viruses described in Czech isolates of *Armillaria* (Figure 1) are hosted by more than one isolate, and even by different species, suggesting interspecies transmission. AALV1 is present in isolate 1 (*Armillaria cepistipes*) and in isolate 7 (*A. ostoyae*), AoALV2 was detected just in isolate 7 (*A. ostoyae*), AoALV3 is present in isolates 7 and 13 (*A. ostoyae*), AoALV4 in isolates 6, 7, 8 and 13 (*A. ostoyae*), and AoTV1 was detected just in isolate 13 (*A. ostoyae*; Appendix A). Interestingly, isolates 1 and 7, between which virus interspecies transmission has occurred, were collected at localities approximately 50 km apart. These results suggest that ambiviruses are transmitted efficiently in *Armillaria* in the Czech Republic, as seems to happen in other regions of the Northern hemisphere [13,24], where ambi-like viruses have been found to be very common in *Armillaria* isolates and occur often in *A. borealis* and *A. mellea* from Finland, Siberia, and Switzerland. This efficiency could also be related to the fact that members of the genus *Armillaria*, as well as other fungi causing tree root rot, such as *Heterobasidion* spp. or *Rosellinia necatrix*, usually form large clonal individuals that grow for decades. It can therefore be considered that virus accumulation and interspecies transmission of mycoviruses are considered rather rare. However, laboratory experiments have shown that these are possible within the fungal genera *Aspergillus* [67], *Sclerotinia* [68] and *Cryphonectria* [69]. In vitro studies have shown mycovirus transmissions between somatically incompatible fungal strains of *Heterobasidion* [70,71,72], being relatively common in both the laboratory and in nature. 

## 4. Conclusions

Our study confirms the presence of one tymovirus and five “ambiviruses” infecting Czech populations of *A. ostoyae* and *A. cepistipes*. Their potential effect on infected *Armillaria* hosts and their ability to be transmitted intra- and interspecies should be further investigated.

## Figures and Tables

**Figure 1 viruses-16-00610-f001:**
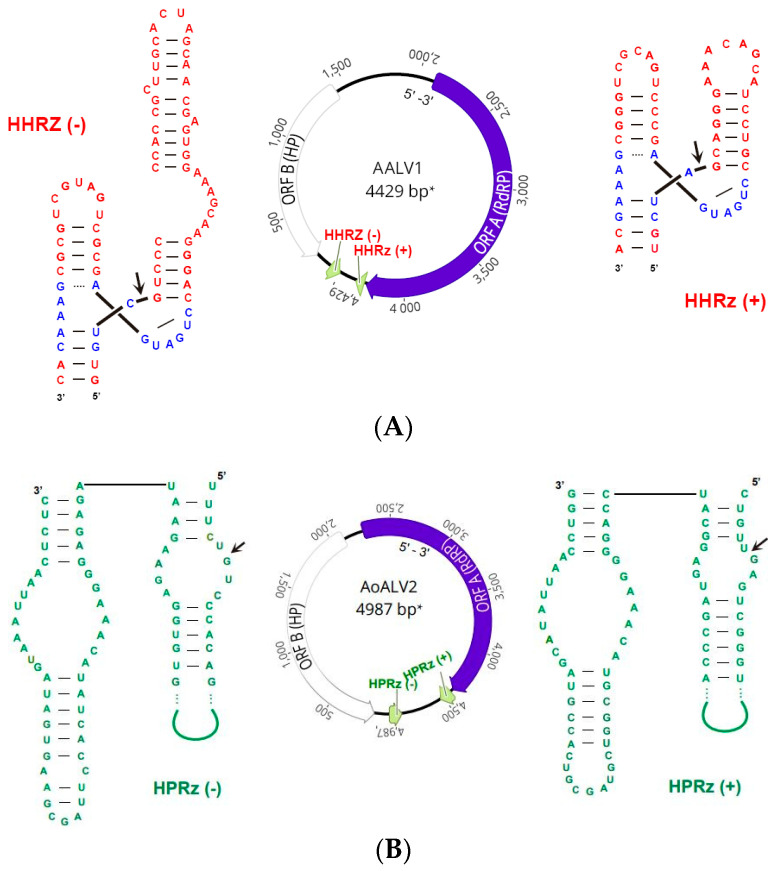
Schematic presentation of AALV1 (**A**), AoALV2 (**B**), AoALV3 (**C**) and AoALV4 (**D**) genomes with predicted ORFs and ribozymes (rbz) secondary structures drawn as inlay; black arrows show the potential self-cleavage site at the predicted HHRz and HPRz motifs. Conserved residues are highlighted in blue; (**E**) schematic representation of AoTV1 genome with predicted ORF and its conserved motifs. The rbz (+) polarity is defined as the RNA strand coding for the polymerase (ORF A). * nt positions of the region spanning the ribozyme in the original virus contig. For more information of the rbz position refer to Appendix A.

**Figure 2 viruses-16-00610-f002:**
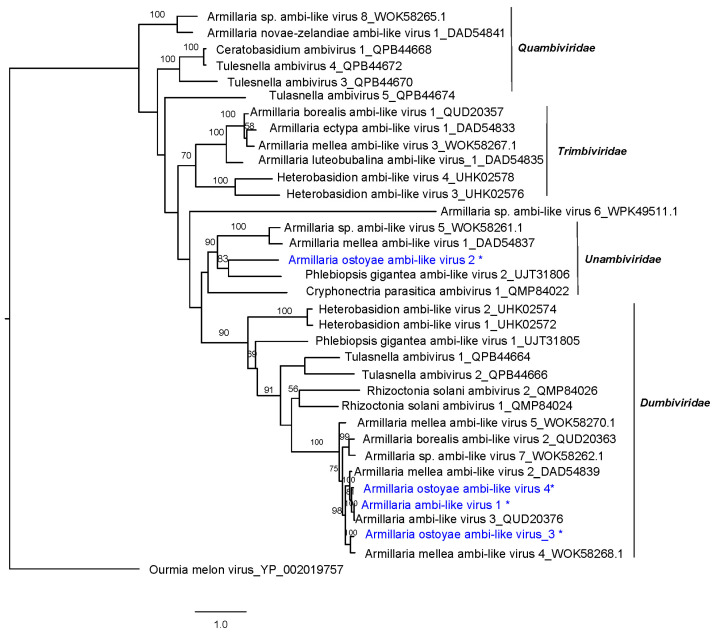
RAxML phylogenetic tree based on the predicted RdRP of representative ambiviruses. Nodes are labeled with bootstrap support values ≥50%. Branch lengths are scaled to the expected underlying number of amino acid substitutions per site. Tree is rooted in the midpoint and uses Ourmia melon virus as an outgroup. Czech *Armillaria* viruses are written in blue and indicated with an asterisk.

**Figure 3 viruses-16-00610-f003:**
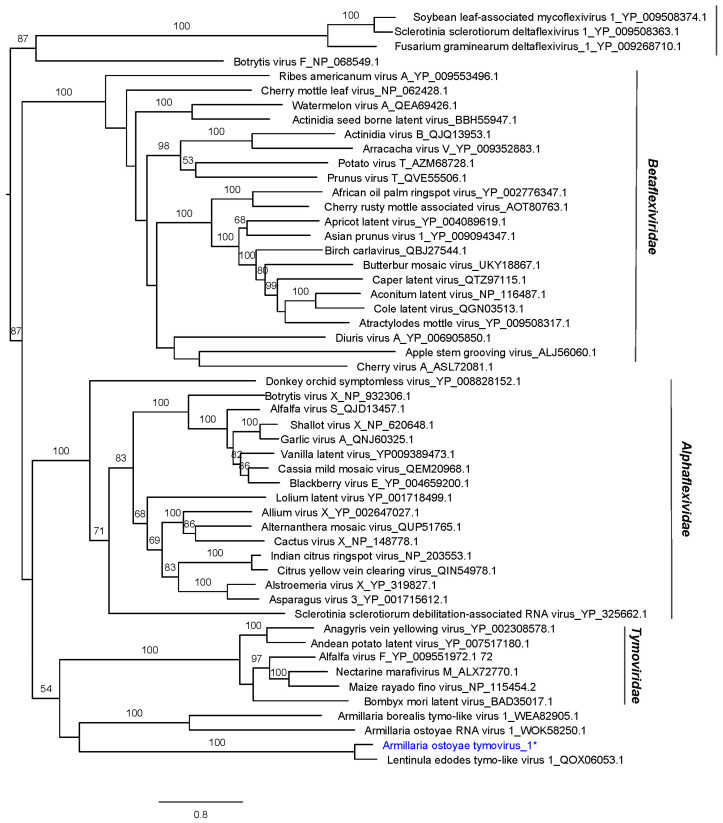
RAxML phylogenetic tree based on the predicted RdRP of Armillaria ostoyae tymovirus 1 (blue letters and *) and other members of the order *Tymovirales*. Nodes are labeled with bootstrap support values ≥50%. Branch lengths are scaled to the expected underlying number of amino acid substitutions per site.

**Table 1 viruses-16-00610-t001:** Data collection of the examined *Armillaria* isolates in this study.

Isolate Code	Fungal Species	Site Coordinates	Country	Tree Host	Fungal Material	Collection Date (Day.Month.Year)
1	*A. cepistipes*	49.5147885 N, 17.5546327 E	Czech Republic	*Picea abies*	Rhizomorphs	29.9.2020
2	*A. ostoyae*	48.9813906 N, 14.4205031 E	Czech Republic	*Picea abies*	Fruiting body	31.10.2019
3	*A. ostoyae*	49.81582 N, 17.34951 E	Czech Republic	*Picea abies*	Fruiting body	27.10.2019
4	*A. ostoyae*	48.6232622 N, 14.6442581 E	Czech Republic	*Picea abies*	Fruiting body	16.10.2019
5	*A. ostoyae*	50.10238 N, 16.0683 E	Czech Republic	*Picea abies*	Fruiting body	23.10.2019
6	*A. ostoyae*	49.5226228 N, 17.5717912 E	Czech Republic	*Picea abies*	Rhizomorphs	5.9.2019
7	*A. ostoyae*	49.8084936 N, 17.4939458 E	Czech Republic	*Picea abies*	Rhizomorphs	4.8.2020
8	*A. ostoyae*	49.31372 N, 16.77221 E	Czech Republic	*Picea abies*	Fruiting body	23.10.2019
9	*A. ostoyae*	48.6078636 N, 14.6688486 E	Czech Republic	*Picea abies*	Fruiting body	21.10.2019
10	*A. ostoyae*	49.8084936 N, 17.4939458 E	Czech Republic	*Picea abies*	Rhizomorphs	4.8.2020
11	*A. ostoyae*	49.32247 N, 16.78645 E	Czech Republic	*Picea abies*	Fruiting body	23.10.2019
12	*A. ostoyae*	48.9815597 N, 14.4162544 E	Czech Republic	*Picea abies*	Fruiting body	4.11.2020
13	*A. ostoyae*	49.5419285 N, 17.3919515 E	Czech Republic	*Picea abies*	Rhizomorphs	14.8.2019

**Table 2 viruses-16-00610-t002:** Mycoviruses detected in *Armillaria* samples in this study.

Virus Name	Acronym	L	Accession Number ^a^	Most Similar Virus ^b^	E Value	Q (%)	I (%)	Mapped Reads	Depth of Coverage
Armillaria ambi-like virus 1	AALV1	4663	ON380550	Armillaria spp. ambi-like virus 3	0.0	96	96.31	24,902	780.91
Armillaria ostoyae ambi-like virus 2	AoALV2	4541	ON380551	Phlebiopsis gigantea ambi-like virus 2	2 × 10^−103^	41	36.65	27,239	863.10
Armillaria ostoyae ambi-like virus 3	AoALV3	4562	ON380552	Armillaria mellea ambi like virus 4	0.0	56	80.94	21,362	678.90
Armillaria ostoyae ambi-like virus 4	AoALV4	4549	ON380553	Armillaria ambi-like virus 3	0.0	46	91.57	6169	196.26
Armillaria ostoyae tymovirus 1	AoTV1	6824	ON380554	Lentinula edodes tymo-like virus 1	0.0	94	68.16	4959	106.22

^a^ Accession number in GenBank. ^b^ Most similar viruses in GenBank (BLASTX) accession numbers: MW423812.1 (Armillaria ambi-like virus 3), MZ448625.1 (Phlebiopsis gigantea ambi-like virus 2), BK014421.1 (Armillaria mellea ambi-like virus 2), MW423813.1 (Armillaria ambi-like virus 3), MN744726.1 (Lentinula edodes tymo-like virus 1); Q, query cover; I, Identity; L, virus sequence length; depth of coverage was calculated by following formula: (Total reads mapped to the final identified virus * average read length)/virus genome or contig length.

## Data Availability

All data generated or analyzed during this study are included in this published article and its Appendix A files.

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
