# Peer review of "Characterization of Mycoviruses in Armillaria ostoyae and A. cepistipes in the Czech Republic"

_viruses, 2024, doi:10.3390/v16040610_

Round 1

Reviewer 1 Report

Comments and Suggestions for Authors

This manuscript is about viruses isolated from different species of pathogenic fungus Armillaria. This study may help in understanding ambivirus evolution. However, several issues need to be addressed before its publication. I would advise authors to be cautious with how they write virus names. The paper requires a moderate language review.

Comment 1 “Introduction”

Here authors described mostly about host fungus and limited information about viruses associated to Armillaria. To fully support the proposed title, "Molecular detection and characterization of mycoviruses in the genus Armillaria in the Czech Republic" it is advisable to incorporate detailed information concerning mycoviruses associated specifically with Armillaria.

The other concerns are listed below.

Line 19 to 22: I recommend authors to restructure the sentence for better clarity.

Line 40: Incomplete sentence, not clear.

Line 41: change “rod” to “rot”.

Line 41: after [12] Is it “ . ” Or  “ , ” ?

Line 40: after [13] Is it “ , ” Or  “ . ” ?

Line 48: Change “Chryphonectria” to “Cryphonectria

Line 64: should be “including”

Line 70: Mention the speed and/or time required for homogenizing the mycelia using vortex.

Line 80: Please mention kind of vortex machine used (e.g. company name/source from)

Line 265 to 268: This is confusing and not clear. Rephrasing the sentence may help readers to understand it better.

Line 299: Please confirm if its “AoALV1” or  “AALV1”.

Line 346: Is it “ wide range of host” or “wide range of virus” or  “wide range of fungal host”? Please clarify.

Line 407: Authors can show the result of RT-PCR in supplementary data, otherwise it is unclear.

Table 2 : May be AoTV1?

Figure 1 (A): Is it “AoALV1” or  “AALV1”. Please confirm.

Figure 2 “phylogenetic tree” can be replaced with better version.

Comments on the Quality of English Language

The paper requires a moderate language review.

Reviewer 2 Report

Comments and Suggestions for Authors

This study conducted screening in search of mycoviruses in 13 isolates of Armillaria spp. collected from the Czech Republic. Five distinct mycoviruses from the Tymoviridae and Ambiviridae families were discovered in these fungi. The manuscript is well-written and clearly describes all the methods and results. Therefore, I recommend that this manuscript be accepted for publication after the authors agree to make some minor edits to it.

Minor comments:

Line 18: What is stranded total RNA-seq?

Line 167: Make it Reverse transcription polymerase chain reaction (RT-PCR)  and Sanger Sequencing.

Line 169: Please mention the composition of lysis buffer.

Line 262: should be Ambivirus?

Comments on the Quality of English Language

I have no issues with the English language, but there are some typos that authors should rectify.

Reviewer 3 Report

Comments and Suggestions for Authors

Lucie et al. conducted molecular identification and characterization of mycoviruses in Armillaria cepistipes and A. ostoyae from the Czech Republic. The authors identified five single-stranded RNA viruses in A. ostoyae and A. cepistipes, including a tymovirus infecting A. ostoyae and four ambivirus-like viruses with circular ambisense genome organization infecting A. cepistipes. Additionally, ribozymes (hammerhead (HHRz) and hairpin (HpRz)) were detected in all the ambiviricot sequences. This brief report will likely enhance our understanding of virus evolution and diversity. However, the manuscript contains several mistakes that require attention and verification. Although the methodology is well-detailed, the authors must rectify the errors. The following are some examples, though the list is not exhaustive.

Title: The title should be clear and concise. Instead of referring to the genus Armillaria, specifying the two species utilized in this study would be preferable.

Abstract:

Lines 20-21: In the results section, you stated that it was the second reported tymovirus, but here you referred to it as the “first”?

Lines 24-26: How can one conclude the “occurrence of interspecies transmission” solely through phylogenetic comparison without conducting any transmission assay or analysis? Viruses can infect two host organisms without necessarily involving interspecies transmission.

Materials and methods:

Lines 60-65: It would be better if the authors could specify the preservation method used for the collected isolates, considering they were collected in 2019-2020.

Lines 70-71: Which protocol? Clarify

Lines 116-117: Despite A. ostoyae being prevalent among the collected isolates, why did you choose not to use the genomes of both species as reference sequences?

Line 141: “evalue”, “e-value”, “E-value”! Be consistent with E-value throughout the manuscript.

Line 163: space between “RNAmotif”

Line 168: What does it mean “a homogenized”?

Line 173: in or at -80 degrees?

Lines 174-175: What does it mean as by Tonka?

Line 176: What does regiontranslation mean?

Lines 178-182: Is it one sentence or two?

Results and discussion:

Line 206: Please specify the species isolated throughout your manuscript.

Line 207: Italicize Tymoviridae

Line 227: What does it mean final ssRNA?

Line 262: Ambivir or ambivirus? RdRp instead of RdRP

Line 267: The full name should be written the first time, followed by the abbreviation used throughout the manuscript.

Line 304: Italicize A. ostoyae

Line 304: Tymoviruses shouldn’t be italicized.

Line 401: You mentioned that it is the second described, but in the abstract, you stated it as the first!

Line 402: 2002 or 2024

Lines 408-426: You cannot draw conclusions based solely on speculation. Without conducting transmission assays and analysis, how can one assert that an interspecies transmission occurred?

Comments on the Quality of English Language

The manuscript contained several typos that need to be addressed.

Reviewer 4 Report

Comments and Suggestions for Authors

The authors detected and characterized the mycoviruses from samples in the genus Armillaria that were collected in the Czech Republic. The result in this manuscript presented the existence of mycoviruses from family Tymoviridae and "Ambiviruses" in the Armillaria sp. This study provides important source for future studies on the mycoviruses-fungi interactions and for the control of fungal pathogen. This manuscript can be considered to publish in viruses until remaining issues are all addressed.

Major issues:

 (1) P3, L85: when the authors conducted the standard total RNA sequencing, did you build only one RNA library with mixed total RNA from 13 samples or 13 libraries for each sample? please clarify it;

(2) P5, L209-211: Please show the result/image of your dsRNA detection; I don’t understand why the dsRNA of viruses couldn't be detected. When ssRNA viruses replicate in host cells, they form dsRNA intermediates which are likely to be isolated and detected. I am curious if these mycoviruses can actively replicate in the collected samples.

 (3) P10, L407-414: PCR analysis data should be included in this manuscript

(4) in the abstract, the authors mentioned two methods they used to detect viruses. However, they couldn’t isolate the dsRNA from collected samples. As the first major method, the authors should include the gel electrophoresis result in the supplementary file at least.

Minor issues:

·       P4, L131: performer-performed;

·       P7, L262: Ambivir- Ambivirus;

·       P7, L271: Figure1, a, b, c, d- Figure1, A, B, C, D;

·       P8, L299: Figure1, A: AoALV1-AALV1, also check the figure;

·       P9, L344: Figure3-Figure1, E;

·       P10, L403: Tymoviridae-Tymoviriale;

·       P8, L340 and P9, L387: please show what region or full genomic sequence was based to build these two phylogenetic tree in this figure legend.

Comments on the Quality of English Language

-

Round 2

Reviewer 4 Report

Comments and Suggestions for Authors

The new version is good!

One suggestion. Although sometimes samples yield negative results, it is still important to keep these outcomes with both positive and negative controls. Doing so helps substantiate the authenticity of the results and demonstrates the feasibility of the method employed.